# Mapping the Neural Substrates of Cocaine Craving: A Systematic Review

**DOI:** 10.3390/brainsci14040329

**Published:** 2024-03-29

**Authors:** Letícia Silvestri Paludetto, Luiza Larrubia Alvares Florence, Julio Torales, Antonio Ventriglio, João Maurício Castaldelli-Maia

**Affiliations:** 1Medical School, FMABC University Center, Santo André 09060-870, Brazil; silvesdetto@gmail.com; 2Department of Preventive Medicine, School of Medicine, Federal University of São Paulo (UNIFESP), São Paulo 04021-001, Brazil; luiza.florence@yahoo.com; 3Department of Medical Psychology, School of Medical Sciences, Universidad Nacional de Asunción, San Lorenzo 111421, Paraguay; juliotorales@gmail.com; 4Regional Institute for Health Research, Universidad Nacional de Caaguazú, Coronel Oviedo 050106, Paraguay; 5School of Medical Sciences, Universidad Sudamericana, Salto del Guairá 130112, Paraguay; 6Department of Clinical and Experimental Medicine, University of Foggia, 71122 Foggia, Italy; a.ventriglio@libero.it; 7Department of Neuroscience, School of Medicine, FMABC University Center, Santo André 09060-870, Brazil; 8Perdizes Institute of the Clinical Hospital (IPer-HCFMUSP), School of Medicine, University of São Paulo, São Paulo 01246-903, Brazil

**Keywords:** cocaine, craving, neurobiology, neuroimaging

## Abstract

Craving is one of the most important symptoms of cocaine use disorder (CUD) since it contributes to the relapse and persistence of such disorder. This systematic review aimed to investigate which brain regions are modulated during cocaine craving. The articles were obtained through searches in the Google Scholar, Regional BVS Portal, PubMed, and Scielo databases. Overall, there was a selection of 36 studies with 1574 individuals, the majority being participants with CUD, whereby about 61.56% were individuals with CUD and 38.44% were controls (mean age = 40.4 years). Besides the methodological points, the neurobiological investigations comprised fMRI (58.34%) and PET (38.89%). The induction of cocaine craving was studied using different methods: exposure to cocaine cues (69.45%), stressful stimuli, food cues, and methylphenidate. Brain activations demonstrated widespread activity across the frontal, parietal, temporal, and occipital lobes, basal ganglia, diencephalon, brainstem, and the limbic system. In addition to abnormalities in prefrontal cortex activity, abnormalities in various other brain regions’ activity contribute to the elucidation of the neurobiology of cocaine craving. Abnormalities in brain activity are justified not only by the dysfunction of dopaminergic pathways but also of the glutamatergic and noradrenergic pathways, and distinct ways of inducing craving demonstrated the involvement of distinct brain circuits and regions.

## 1. Introduction

Cocaine use disorder (CUD) has a chronic course, multifactorial causes, and significant prevalence [1]. Moreover, it is associated with severe impairment of the emotional, social, and physical integrity of users due to infectious diseases such as HIV and hepatitis C [1]; acute intoxication; craving; withdrawal syndrome; overdose; and death due to related complications, such as stroke, coronary artery disease, and acute myocardial infarction [2]. It is estimated that globally, around 21.5 million individuals used cocaine at least once in 2020, corresponding to about 0.4% of the world population aged 15 to 64 years old [3]. Although this data pertains to cocaine use, it is known that a portion of these users will develop CUD and have to cope with its various consequences, including craving.

Craving is defined as a strong desire or impulse to use a substance [4] and results from neuroadaptations in mesocorticolimbic and nigrostriatal dopaminergic pathways [4,5]. These neuroadaptations are progressive and contribute to the transition from controlled and occasional use to abuse and, consequently, addiction [4] by interfering with the ability to exert self-control over substance consumption and increasing the sensitivity to stress and negative feelings [5,6].

As CUD presents significant individual impairments and craving contributes to relapse [7,8,9,10,11,12,13,14,15,16,17,18], clarifying its neurobiology to develop support and treatment strategies is crucial. Therefore, the present review aims to investigate which brain regions are modulated during cocaine craving. 

Substance use disorders are explained by a model that divides addiction into a cycle with three stages: (1) binge/intoxication, (2) withdrawal/negative affect, and (3) preoccupation/anticipation [4]. Each of these stages is associated with the activation of specific brain circuits and regions that contribute to the development and persistence of these disorders [4,19]. Here, we focus on the third stage of addiction.

The preoccupation/anticipation stage in substance use disorders is linked to compromised prefrontal cortex (PFC) activity, leading to craving and substance-seeking behavior [4,20]. PFC activity is modulated by two interconnected systems: the “Go system” and the “Stop system” [4,20]. The “Go system” is responsible for decision-making, planning, and initiating goal-directed behaviors, often influenced by habits via interactions with the basal ganglia (BG) [4,20]. Key components of the “Go system” include the anterior cingulate cortex (ACC), dorsal PFC, orbitofrontal cortex (OFC), nucleus accumbens (NAc), and ventral tegmental area (VTA) [20]. Conversely, the “Stop system” regulates the “Go system” and is associated with emotions and stress-related circuits and brain regions [4]. It encompasses the ventral PFC and other regions that intersect with the “Go system” [20].

Substance-seeking behavior in substance use disorders can be triggered by exposure to the substance or substance-related cues (“reward craving”) or by exposure to stress-inducing stimuli (“relief craving”) [20]. Dysregulation of the “Stop system” is associated with both types of craving [4]. When individuals are exposed to the substance or substance-related cues, the “Go system” becomes highly active as the “Stop system” fails to regulate it, leading to craving and intensified substance use. This is mediated by glutamatergic interactions with the BG, particularly the NAc and the dorsal striatum (DS), which are critical in the binge/intoxication stage [4,19].

## 2. Materials and Methods

### 2.1. Eligibility

Original articles published in English until January 2023, addressing cocaine craving using imaging techniques and neurostimulation/modulation methods were selected from search results in four databases and organized into an XLSX format table (Appendix A). 

### 2.2. Databases

The articles were obtained through searches in the Google Scholar, Regional BVS Portal, PubMed, and Scielo databases. The databases were chosen based on their accessibility (whether institutional access was granted to the first author when needed). Google Scholar served as a valuable resource for accessing gray literature. Its comprehensive coverage includes a wide range of sources, such as theses, dissertations, conference papers, and preprints, making it an indispensable tool to explore non-traditional scholarly materials.

### 2.3. Search

The research was conducted according to the specific search functionalities of each database. The keywords used included (1) terms referring to neurobiological investigation methods (“functional magnetic resonance imaging,” “fMRI,” “magnetoencephalography,” “MEG,” “near-infrared spectroscopy,” “NIRS,” “positron emission tomography,” “PET,” “single-unit recording,” “transcranial direct-current stimulation,” “tDCS,” “transcranial magnetic stimulation,” and “TMS”) and (2) terms related to the investigated phenomenon (“craving” and “cocaine”).

### 2.4. Article Selection

The first author conducted the entire article selection process in accordance with the Preferred Reporting Items for Systematic Reviews and Meta-Analyses (PRISMA). The flowchart illustrates the identification, screening, eligibility, and inclusion of articles, outlining the steps of this process leading to the final article selection (Figure 1). The records were screened by title and abstract (Appendix A) by the first author and supervised by the last author. Disagreements were resolved by discussion with the second author. 

### 2.5. Data Collection

The entire data collection process was conducted by the first author, assisted by the supervisor. The first author read the 36 articles that comprised the final selection, and the collected data were organized into an XLSX format table (Appendix A).

### 2.6. Data Division

The present study considered the following variables: number of participants, age, biological sex, cocaine consumption status, methodology, main findings, and limitations.

### 2.7. Register

The protocol of the present study was registered on the Open Science Framework (OSF) on 3 August 2023, and can be accessed through the link https://osf.io/ahsu5.

## 3. Results

### 3.1. Sample Characteristics

The final selection of 36 studies included a total of 1574 participants, consisting of 969 individuals with CUD (61.56%) and 605 controls (38.44%). The mean age of the participants was 40.4 years, with a standard deviation (SD) of 6.6 years [10,11,13,14,18,21,22,23,24,25,26,27,28,29,30,31,32,33,34,35,36,37,38]. 

### 3.2. Cocaine Use Status

The cocaine use status of individuals with CUD is highly heterogeneous, with some individuals actively using cocaine and others in abstinence (hours to weeks); however, in all cases, cocaine was the drug of choice. In 19 studies (52.78%) [10,11,12,13,14,18,21,22,23,24,25,26,27,28,29,30,31,39,40], individuals with CUD were in various stages of abstinence; in 9 studies (25%) [9,15,16,32,33,34,41,42,43], individuals with CUD were in various stages of abstinence and seeking treatment or were in treatment; in 2 studies (5.56%) [35,44], individuals with CUD were not in abstinence but were seeking treatment; in 6 studies (16.68%) [7,8,17,36,37,38], individuals with CUD were neither in abstinence nor seeking treatment. Additionally, in 25 studies (69.45%) [9,10,12,13,15,16,17,18,22,23,24,26,28,29,30,31,32,34,35,36,37,38,40,41,43], participants used, in addition to cocaine, alcohol and/or nicotine and/or caffeine.

### 3.3. Metodology

In 21 studies (58.34%) [7,9,11,12,15,16,17,18,23,25,26,28,29,30,31,34,36,39,40,43,44], the neurobiological investigation method used was Fmri; in 14 studies (38.89%) [10,13,14,21,22,24,27,32,33,35,37,38,41,42], the neurobiological investigation method used was PET; and in 1 study (2.78%) [8], the neurobiological method used was morphometric analysis. In 15 studies (41.67%) [7,10,13,14,17,21,26,27,33,34,37,40,42,44], cocaine craving was induced by exposure to cocaine cues; in 2 studies (5.56%) [9,39], it was induced by exposure to stressful stimuli; in 3 studies (8.34%) [11,15,43], it was induced both by cocaine cues and stressful stimuli; in 2 studies (5.56%) [25,29], it was induced both by cocaine cues and food cues; in 3 studies (8.34%) [32,35,41], it was induced by methylphenidate (MP); in 1 study (2.78%) [24], it was induced both by cocaine cues and MP; in 2 studies (5.56%) [16,18], it was induced by monetary rewards; and in 1 study (2.78%) [23], it was induced by cocaine itself.

### 3.4. Main Findings

#### 3.4.1. Brain Activation

Findings indicate diffuse brain activation. Although craving is primarily attributed to impaired PFC activity [4], it also depends on brain regions in the parietal, temporal, and occipital lobes, as well as the diencephalon, brainstem, and limbic system.

##### Frontal Lobe

Activations in the frontal lobe include activations of the frontal cortex [17,26,36], bilateral middle frontal gyrus (MFG) [29,34,40], motor cortex [40], and PFC [7,16,17,21,25,26,32,35,43]. More specifically, activations in the frontal lobe include activations of the supplementary motor area (SMA) [40]; bilateral medial frontal cortex (MFC) [26,36]; bilateral superior frontal cortex [17]; right [25,26,32] and left [17,25,26] OFC [17,21,25,26,32]; right orbital and medial prefrontal cortex (OMPFC) [35]; left [26] dorsolateral prefrontal cortex (dlPFC) [25,26,43]; dorsomedial prefrontal cortex (dmPFC) [17,43]; ventrolateral prefrontal cortex (vlPFC) [43]; and ventromedial prefrontal cortex (vmPFC) [17,43].

##### Parietal, Temporal, and Occipital Lobes

Activations in the parietal lobe include activations of the left [29] inferior (IPC) [29,34] parietal cortex [29,34,36,43], bilateral inferior parietal gyrus [29,40], left inferior parietal lobe [16], right inferior parietal lobule (IPL) [7], and precuneus (PCu) [11].

Activations in the temporal lobe [42] include activations of the bilateral [36] temporal cortex [36,43] and the left superior temporal gyrus [12].

Activations in the occipital lobe include activations of the right [17], left [25], and bilateral [36] occipital cortex [13,17,25,36]; bilateral visual cortex [29,34,40]; and left calcarine sulcus [16]. It is worth noting the activation of the left fusiform gyrus [16], which occupies both the temporal and occipital lobes.

##### Basal Ganglia, Diencephalon, and Brainstem

Activations in the basal ganglia include activations of the striatum (ST) [7,16,17,18,26,32,43]. Specifically, they include activations of the DS [7,16,18,26], right [26] VS [17,18,26,43], and globus pallidus [18]. Activations in the DS include activations of the right [18] and left [7,18] caudate nucleus and the right [16,18] and left [18] putamen. Activations in the ventral striatum (VS) include activations of bilateral Nac [17].

Activations in the diencephalon include activations of the hypothalamus [15,29,34,40] and the right [11], left [12], and anterior [25] thalamus [11,12,15,25,35,40]. Activations in the cerebellum [12,15,21,32,36] include activations in the right [12,32] and left [32] cerebellum.

Activations in the brainstem include activations in the ventral tegmental area (VTA) [15], left pretectal area [12], periaqueductal gray (PAG) [15,40], and locus coeruleus (LC) [15].

##### Limbic System

Activations in the limbic system include activations of the right [15,26] and left [26] amygdala [15,25,26,42]; anterior [7], posterior [16], and superior [32] cingulate gyrus [7,16,32]; anterior (ACC) [11,26,42,43,44] and posterior (PCC) [11,43] cingulate cortex (CC) [11,26,42,43,44]; right [17,26] and left [17] hippocampus [15,16,17,26]; parahippocampal gyrus (PHG) [25]; and right [26] and left [11,21,26] insula [11,12,21,26,43].

## 4. Discussion

The present review aimed to investigate which brain regions are modulated during cocaine craving. It is known that craving is essentially associated with impaired PFC activity [4,19,20]. Therefore, as expected, studies showed impairment of PFC activity and the correlation between such impairment and craving in individuals with CUD (Figure 2). However, they also showed abnormalities in the activity of various other brain circuits and regions and the correlation between such abnormalities and craving, which reinforce, complement, and/or modify the current understanding of the neurobiology of craving (Figure 2 and Figure 3). The variety of results presented in the current study may not lead to definitive conclusions; however, it is important to emphasize the contribution of each of the studies to the neurobiology of cocaine craving.

### 4.1. Impairment of Frontal Lobe Activity

The impairment of PFC activity is associated with difficulties in controlling executive functions, decision-making, and emotional regulation in individuals with CUD [4,19]. This impairment often results from an imbalance between the “Go system” and “Stop system” or dysfunction of the “Stop system,” leading to the Impaired Response Inhibition and Salience Attribution (iRISA) syndrome [46]. The iRISA syndrome is characterized by decreased inhibition of harmful behaviors, increased salience attribution to substances and associated stimuli, and reduced sensitivity to natural stimuli. These factors contribute to compulsive drug-seeking and drug-taking behaviors [46].

Individuals with CUD often exhibit increased stop signal reaction time (SSRT) and decreased post-signal slowing (PSS) during a stop signal task, indicating difficulties in response inhibition and PFC dysfunction [47]. Additionally, reduced activation of the ACC further highlights the challenges of impulse control in this population [45].

Despite the automatic nature of craving, studies have shown that individuals with CUD can inhibit craving when instructed to do so, albeit at the cost of decreased activity in the right NAc and mOFC, mediated by the right IFC [13]. Strengthening connectivity between dopaminergic projections from the VTA and glutamatergic projections from the PFC to modulate the NAc and mOFC responses to cocaine cues is a potential therapeutic strategy to enhance executive inhibitory capacity in individuals with CUD.

The primary motor cortex, including the PCG, plays a role in executive functions, particularly in the development of automated actions. Studies have demonstrated PCG activation in response to cocaine cues and its correlation with craving and relapse [7,48]. Recent research has also shown PCG activation in response to monetary rewards, with tonic cocaine craving mediating the relationship between days of cocaine use and PCG activation, suggesting PCG activation as a marker of tonic cocaine craving [18].

While addictive substances, especially stimulants, typically increase dopamine (DA) in the reward circuit, recent studies have indicated DA release in the PFC in response to cocaine cues and MP [27,32]. This release of DA in the PFC reflects communication between subcortical and cortical regions through the cortico-striato-thalamo-cortical circuit. Chronic substance use-induced DA release in the PFC leads to alterations in meso-fronto-striatal connectivity and subsequent changes in PFC activity. Increased OFC-VS connectivity is associated with heightened salience attribution to cocaine cues, negatively correlating with the PCu and PCC. These changes contribute to impaired decision-making and reduced craving resistance, ultimately facilitating craving in response to cocaine cues and promoting compulsive substance-seeking behavior [25,27].

Additionally, DA release in specific PFC regions, including the vmPFC and dlPFC, is positively correlated with the intensity of craving in response to cocaine cues [27]. However, increased DA alone is insufficient to activate the frontal lobe. Activation of the PFC occurs selectively in individuals with CUD when MP enhances mood, while OFC activation is observed when MP induces craving. OFC activation is associated with processing salient stimuli and anticipating a stimulus, highlighting its role in the perception of reinforcing effects and expectations of receiving another dose of MP [32].

### 4.2. Impairment of the Activity of other Brain Regions 

#### 4.2.1. Basal Ganglia

Subcortical brain structures, including the ST, VS, and DS, are essential for various cognitive functions. The ST is involved in motor planning, decision-making, motivation, and reward processing [49]. The VS contributes to reward processing and responds to negative emotions and stimuli such as pain [31]. Conversely, the DS plays a role in action selection, initiation, and habit formation [4,10]. It is particularly crucial during the development of compulsive substance-seeking behavior, facilitating the transition to compulsive cocaine use [50].

Milella et al. demonstrated DA release in the ST upon exposure to cocaine cues, with a direct correlation between the magnitude of DA release and the availability of D2 receptors in the mesencephalon. However, the temporal relationship between mesencephalic D2 receptor availability and substance use remains unclear [27]. Volkow et al. revealed DA release in the DS, specifically in the caudate nucleus and putamen, upon exposure to cocaine cues, directly correlating with craving. This release occurs when seeking the substance is necessary for acquisition, underscoring the DS’s role in mediating compulsive substance-seeking behavior, action selection, initiation, and habit formation [10]. The magnitude of DA release in the DS is influenced by the nigrostriatal pathway, with higher DA release corresponding to increased craving, highlighting its role in craving perception [10].

DA release triggered by exposure to cocaine cues is regulated by glutamatergic pathways, such as the frontostriatal and frontomesencephalic pathways [24]. It is noteworthy that slow DA increases resulting from oral MP administration in the absence of concurrent cocaine cues do not induce craving [24]. In essence, while DA release is necessary, it alone is insufficient to elicit craving, with the concurrent presence of cocaine cues being crucial [24]. Consequently, the increased DA availability in the presence of cocaine cues reflects secondary responses to pathways controlling DA release, such as the frontostriatal and frontomesencephalic pathways [24]. Moreover, the rapid increase in DA, rather than the slow increase induced by oral MP administration, appears to be associated with craving [24].

#### 4.2.2. Diencephalon, Cerebellum, and Brain Stem

The thalamus, responsible for filtering and transmitting nerve impulses from the body to specific brain regions, plays a multifaceted role in regulating pain, sensitivity, motor skills, speech, cognition, mood, and motivation [51]. However, its role in substance use disorders remains unclear, as it is not typically activated during studies attempting to induce craving [11]. Volkow et al. demonstrated a correlation between craving and increased thalamic responsiveness to MP, attributed to dysfunction in the dopaminergic regulation of the cortico–striato–thalamic circuit in individuals with substance use disorders [32,41]. The thalamus is sensitive to cocaine’s effects mediated by D2 receptors, but these receptors are reduced in individuals with substance use disorders, leading to metabolic dysfunction in the OFC and dysfunction of the cortico–striato–thalamic circuit. This circuit, along with its connections to the OFC, dlPFC, and CC, is believed to mediate reward circuit dysfunction in substance use disorders [11].

The cerebellum, widely recognized ”or i’s role in motor function, also influences learning unconscious responses, like craving, and conditioned responses, such as impulsive cocaine use [21]. Through its connections to the limbic system, cerebellar activation stimulates structures within the reward circuit, highlighting its involvement in reinforcing stimuli associated with cocaine.

The VTA is crucial for reward-related and addiction-related behaviors, goal-directed actions like substance-seeking behavior, motivation, and reinforcement learning via DA firing. Conversely, the PAG plays a significant role in pain modulation, autonomic responses, and learning, promoting both defensive and aversive behaviors [52]. Xu et al. demonstrated activation of the VTA and PAG in individuals with CUD exposed to stressful stimuli, particularly those with a specific genetic polymorphism. In this scenario, both brain regions, along with the hypothalamus, thalamus, and cerebellum, contribute to an intensified response to stressful stimuli, increasing the risk of relapse. In the VTA, kappa opioid receptors modulate dopaminergic projections to the prefrontal cortex, while in the PAG, they modulate negative emotions experienced during craving, particularly intense in individuals with specific genetic variants when exposed to stress [15].

Notably, increased neuromelanin signal (NMS) in the LC indicates noradrenergic dysfunction resulting from cocaine-induced neurotoxicity to noradrenergic neurons. Interestingly, the absence of NMS in the VTA/SNc suggests a greater vulnerability of LC noradrenergic neurons to cocaine [36].

#### 4.2.3. Limbic System

The amygdala plays a pivotal role in processing emotions related to various stimuli, including aversive stimuli, fear, food, sex, and substances [53]. It is also involved in regulating responses to stress [53]. In contrast, the hippocampus is crucial for memory consolidation, decision-making, and emotional memory formation [54]. Both the amygdala and the hippocampus contribute to the consolidation of emotionally charged memories [53]. Fotros et al. found DA release in the amygdala, hippocampus, and ST in response to cocaine cues, highlighting an integration between the limbic system and the ST that contributes to incentive salience [14]. This DA release was more pronounced in individuals with CUD reporting intense craving, indicating their heightened sensitivity to assigning incentive salience to cocaine cues [14]. In individuals with CUD and a history of childhood trauma, this integration, particularly between the amygdala and the ST, is notably strong during the processing of cocaine cues, and it appears to be linked to habit formation and cognitive impairment, potentially elucidating how childhood trauma contributes to CUD development and prevalence [17]. Another critical integration involves the amygdala, ACC, and Nac, which, when hyperstimulated by cocaine, leads to reinforcement learning [42].

The ACC plays a role In associati”g em’Iions with environmental stimuli, mediating emotional responses to both rewarding and aversive stimuli [35]. ACC activation is observed during exposure to cocaine cues [7]. Volkow et al. demonstrated a correlation between increased ACC activation and heightened emotional responses to MP in individuals with substance use disorders [35]. Due to the extensive connectivity between the amygdala and the ACC, the ACC is believed to participate in the processing of stressful stimuli. Duncan et al. reported ACC activation during exposure to cocaine cues and stressful stimuli, suggesting its involvement in stress-related processes [11]. However, contrary to Duncan et al.’s findings, Sinha et al. showed ACC deactivation during exposure to stressful stimuli, and greater deactivation was associated with increased craving [9].

The PCC also demonstrated activation during exposure to stressful stimuli, potentially playing a role in reward processing influenced by stress. Kilts et al. argued that the PCC specializes in processing rewards [33]. The hypothalamus, responsible for maintaining homeostasis, including hormone synthesis, autonomic function control, and processing pain, is linked to regulating food intake and reward [34]. Zhang et al. identified the hypothalamus as a neural marker for CUD due to its activation during exposure to cocaine cues and its direct correlation with craving [34]. In a subsequent study, Zhang et al. further confirmed these findings and showed increased hypothalamic activation in individuals with CUD exposed to cues related to food, indicating the plasticity of dopaminergic circuits resulting from chronic cocaine use and the abnormal activation of the hypothalamus in response to natural stimuli [29].

ThI insula serves In interoceptive function, connecting autonomic and visceral information with feelings and motivations, with the left insula involved in the retrieval of semantic and procedural memories, while the right insula is associated with memories linked to emotions [21]. Increased activation of the left insula was observed in response to stimuli associated with cocaine use [11,21]. On the other hand, only the right insula showed a significant correlation between its activation and craving, indicating that individuals with CUD reporting craving were more likely to retrieve emotionally charged memories when exposed to cocaine-related stimuli [21]. Activation of the posterior insula, in conjunction with thalamic activation, suggests its potential role as a mediator of craving, particularly when exposed to monetary rewards, highlighting its involvement in connecting salient stimuli and craving [12].

### 4.3. Limitations

The present review has several limitations. Institutional access was not granted to PsycINFO and Web of Science. A large number of results were obtained in Google Scholar, potentially preventing the initial selection of some studies. The screening process was conducted only by one person, reducing the reliability of the final article selection. Additionally, in most studies, participants used alcohol, nicotine, and/or caffeine, and the potential interference of the effects of these substances on the neurobiology of cocaine craving cannot be ignored. Finally, sex differences and laterality in brain activations were not assessed by this review, nor was the impact of abstinence duration on the neurobiology of cocaine craving.

One additional limitation of our systematic review is the variation in the types of stimuli used to induce cocaine cravings across the included studies. While the majority of studies utilized exposure to cocaine cues, some employed different methods, such as stressful stimuli, food cues, and methylphenidate. This heterogeneity in stimulus type may have contributed to differences in the observed brain activations across studies.

### 4.4. Implications and Future Studies

Craving is related not only to dopaminergic dysfunction but also to glutamatergic [19,50] and noradrenergic dysfunctions [36]. Few studies have focused on the glutamatergic dysfunction of cocaine use disorder due to the lack of radiopharmaceuticals to assess glutamate neurotransmission [50]. Thus, craving has been strongly limited to DA, making it challenging to develop support and treatment strategies that encompass all facets of CUD.

When addressing comorbidities in individuals with CUD, it is important to consider their impact on the development and prevalence of CUD, such as in individuals with CUD and a history of childhood trauma [17], as well as on the treatment of CUD, such as in individuals with CUD and attention-deficit/hyperactivity disorder (ADHD) [24]. Additionally, since cocaine distorts the reward and emotion circuits, which are involved in various everyday activities beyond substance use disorders, understanding what is already known about the normal functions of these circuits, such as learning, memory, and emotions, can be applied to both understanding craving and developing and improving support and treatment strategies [7].

Furthermore, despite the present study not focusing on evaluating differences between the female and male sexes, some specificities have been identified [16,33,39,40,43]. Therefore, it is crucial to consider the sex of individuals with CUD during the study of the neurobiology of cocaine craving, during the development and improvement of support and treatment strategies, as well as when choosing the best therapeutic intervention available for individuals with CUD seeking treatment.

Moreover, these results about cocaine craving should be placed in the context of the larger scenarios of stimulant addiction research, and methamphetamine should be one of the substances to be compared to. Even though cocaine and methamphetamine are both potent stimulants with similar mechanisms of action, which involve dopamine release in the brain’s reward circuitry, they could also possess some pharmacokinetic and neurobiological-related peculiarities that may lead to different craving and addiction-related behavior patterns. Past research on methamphetamine addiction has pointed to the involvement of the frontal lobe and the prefrontal cortex in disrupted executive functions, similarly to that of individuals addicted to CUD [55]. Likewise, dysfunction of subcortical structures such as the basal ganglia and limbic system, which includes the amygdala and the hippocampus, but some other structures as well, methamphetamine craving, and compulsive drug-seeking behaviors have been linked [56]. Through looking at cocaine and methamphetamine from the same neurobiological viewpoint, researchers will be able to discover common neural aspects that lead to stimulant addiction and also those specific to the substance that may be used to devise treatment measures.

## 5. Conclusions

In addition to abnormalities in PFC activity, abnormalities in the activity of various other brain regions contribute to the elucidation of the neurobiology of cocaine craving. Brain regions not commonly included in studies attempting to induce craving, such as the primary motor cortex and the thalamus, are particularly important to guide future studies. Abnormalities in brain activity are justified not only by dysfunction of dopaminergic pathways but also glutamatergic and noradrenergic pathways, and distinct ways of inducing craving demonstrated the involvement of distinct brain circuits and regions. Stressful stimuli are directly correlated with a greater risk of relapse; therefore, along with cocaine cues, they play an important role in CUD and should be taken into account during the development and improvement of medications and therapies for CUD, as well as when choosing the best therapeutic intervention available for individuals with CUD seeking treatment

## Figures and Tables

**Figure 1 brainsci-14-00329-f001:**
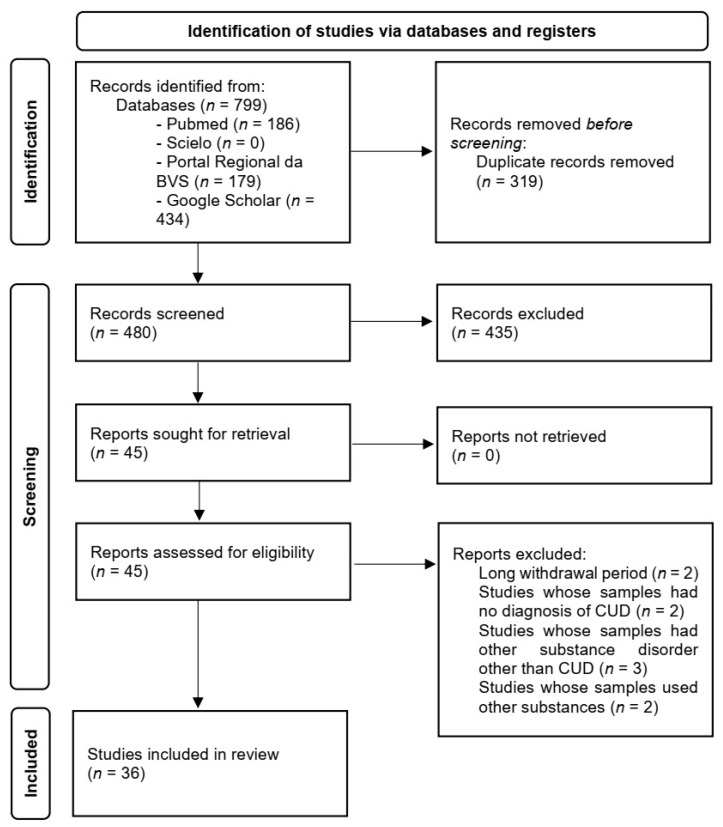
Identification of studies via databases and registers in accordance with the PRISMA 2020 flow diagram for new systematic reviews.

**Figure 2 brainsci-14-00329-f002:**
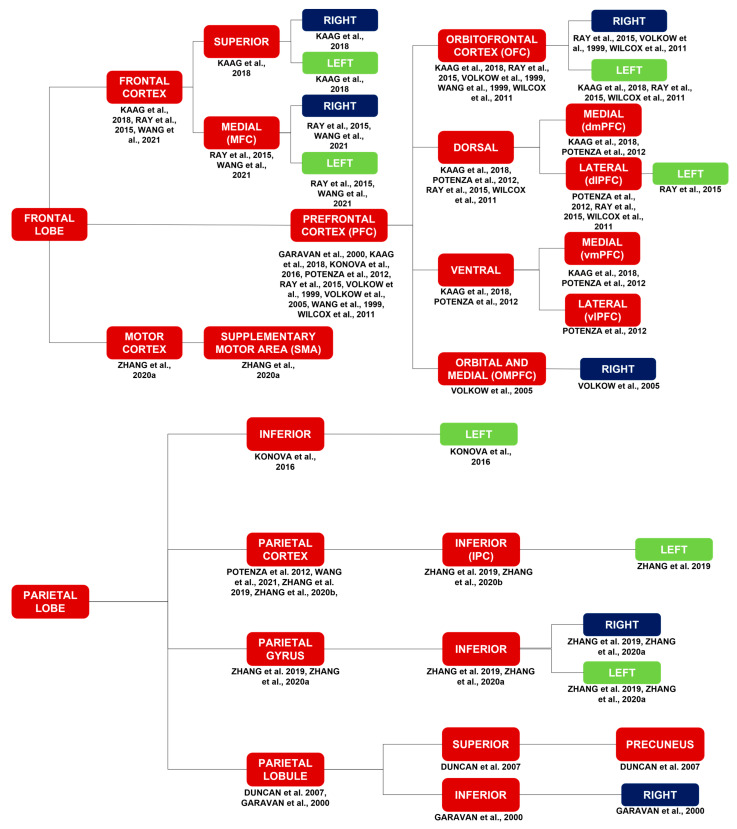
Brain activations in response to a myriad of stimuli, such as cocaine cues, stress, MP, monetary rewards, and cocaine itself. Brain activations in general correspond to the red and purple blocks. Brain activations of the limbic system correspond to the purple blocks. The limbic system is highlighted due to its importance in substance use disorders in general. The laterality of the brain activations corresponds to the blue and green blocks [7,11,12,13,15,16,17,18,21,25,26,29,32,34,35,38,40,42,43].

**Figure 3 brainsci-14-00329-f003:**
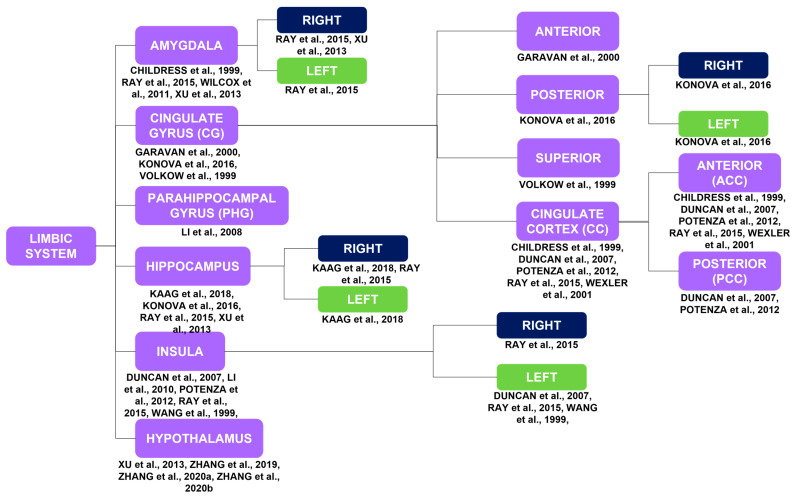
Brain activations of the limbic system in response to a myriad of stimuli, such as cocaine cues, stress, MP, monetary rewards, and cocaine itself, correspond to the purple blocks. The limbic system is highlighted due to its importance in substance use disorders in general. The laterality of the brain activations corresponds to the blue and green blocks [7,11,12,15,16,17,21,25,26,29,32,34,40,42,43,44,45].

## Data Availability

All data obtained from the final article selection and all data generated by the present study are presented in the manuscript and in its Appendix A.

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
