# Peer review of "Mapping the Neural Substrates of Cocaine Craving: A Systematic Review"

_brainsci, 2024, doi:10.3390/brainsci14040329_

Round 1

Reviewer 1 Report

Comments and Suggestions for Authors

The writing style is acceptable, but I have some fundamental issues on the methods of the study as well as some minor points for improvement of the paper:

1. Search strategy and inclusion criteria should be briefly explained in Abstract.

2. Headings of Introduction are not useful enough.

3. Cocaine as a part of stimulants should be discussed in the manuscript as well and the current findings with previous research on other stimulants like methamphetamine should be discussed in Discussion.

4. It should be mentioned if a database covered grey-literature.

5. The manuscript needs exact proof-reading. See line 162 'bilateral lateral...'.

6. Some detailed results which mentioned in Abstract are missed in the text of the manuscript.

7. Colors and numbers in Fig-2&3 should be described clearly.

8. 'Comprehensive' does not make sense for title of this study.

9. Main issue about this review article is lack of conclusive findings. The authors have mentioned several brain regions to be related with craving of cocaine without any conclusive analyses. Without a clear meta-analyses this findings is not so useful for future research.

Comments on the Quality of English Language

Minor editing of English language required.

Author Response

We would like to extend our gratitude to the reviewer for their comprehensive feedback on our manuscript. We have carefully considered each point raised and made appropriate revisions to address the concerns raised. All the changes within the manuscript are highlighted in yellow. Below, we provide a detailed response to each of the reviewer's comments.

Comment:

The writing style is acceptable, but I have some fundamental issues on the methods of the study as well as some minor points for improvement of the paper:

  1. Search strategy and inclusion criteria should be briefly explained in Abstract.

Response:

We have included in the eleven first lines of the methods sub-section of the present manuscript.

Comment:

  1. Headings of Introduction are not useful enough.

Response:

We have excluded such headings from the introduction of the present manuscript.

Comment:

  1. Cocaine as a part of stimulants should be discussed in the manuscript as well and the current findings with previous research on other stimulants like methamphetamine should be discussed in Discussion.

Response:

Following the reviewer’s suggestion, we have included the following paragraph as the last of the one of the section 4.4 of the present manuscript:

“Moreover, these results about cocaine craving should be placed into the context of the larger scenarios of stimulant addiction research; and, methamphetamine will be one of the substances to be compared to. Even though cocaine and methamphetamine are both potent stimulants with similar mechanisms of action, which involve dopamine release in the brain's reward circuitry, they could also possess some pharmacokinetic and neuro-biological-related peculiarities that may lead to different craving and addiction-related behaviors patterns. Past researches on methamphetamine addiction have pointed to the involvement of the frontal lobe and the prefrontal cortex in the disrupted executive functions, similarly to that of individuals addicted to CUD [64]. Likewise, dysfunction of subcortical structures such as the basal ganglia and limbic system, which includes the amygdala and the hippocampus but some other structures as well, methamphetamine craving and compulsive drug-seeking behaviors have been linked [65]. Through looking at cocaine and methamphetamine from the same neurobiological viewpoint, researchers will be able to discover common neural aspects that lead to stimulant addiction and also those specific to the substance which may be used to devise treatment measures.”

  1. Gong, M., Shen, Y., Liang, W., Zhang, Z., He, C., Lou, M., & Xu, Z. (2022). Impairments in the Default Mode and Executive Networks in Methamphetamine Users During Short-Term Abstinence. International Journal of General Medicine, 6073-6084.
  2. Venniro, M., Russell, T. I., Ramsey, L. A., Richie, C. T., Lesscher, H. M., Giovanetti, S. M., ... & Shaham, Y. (2020). Abstinence-dependent dissociable central amygdala microcircuits control drug craving. Proceedings of the National Academy of Sciences, 117(14), 8126-8134.

Comment:

  1. It should be mentioned if a database covered grey-literature.

Response:

We have included the following statement in the end of the section 2.2 of the present manuscript:

“Google Scholar served as a valuable resource for accessing grey literature. Its compre-hensive coverage includes a wide range of sources such as theses, dissertations, conference papers, and preprints, making it an indispensable tool to explore non-traditional scholarly materials.”

Comment:

  1. The manuscript needs exact proof-reading. See line 162 'bilateral lateral...'.

Response:

We have corrected this in the present manuscript.

Comment:

  1. Some detailed results which mentioned in Abstract are missed in the text of the manuscript.

Response:

We have modified the Results sub-section of the Abstract of the present manuscript, as follows:

“Results: Overall, there was selection of 36 studies with 1574 individuals, majority being partici-pants with cocaine use disorder (CUDD) whereby about 61.56% were individuals with CUDD and 38.44% controls with mean age of 40.4 years. With regard to the status of CUD among indi-viduals who encountered CUD, the findings were that some were still using cocaine, like few were in abstinence and others were seeking treatment. Besides the methodological points, the neurobi-ological investigations were comprising fMRI (58.34%) and PET (38.89%). The induction of co-caine craving was studied using different methods: exposure to cocaine cues (69.45%), stressful stimuli, food cues, and methylphenidate. Brain activations demonstrated widespread activity across frontal (e.g., frontal cortex, middle frontal gyrus, prefrontal cortex), parietal (e.g., inferior parietal cortex, precuneus), temporal (e.g., temporal cortex, superior temporal gyrus), and occipi-tal (e.g., occipital cortex, visual cortex) lobes, basal ganglia (e.g., striatum, caudate nucleus), dien-cephalon (e.g., hypothalamus, thalamus), brainstem (e.g., ventral tegmental area, periaqueductal gray), and limbic system (e.g., amygdala, cingulate gyrus, hippocampus, insula).”

Comment:

  1. Colors and numbers in Fig-2&3 should be described clearly.

Response:

We have clarified this in the legends of the Figures 2 and 3, as follows:

“Figure 2: Brain activations in response to a myriad of stimuli, such as cocaine cues, stress, MP, mone-tary rewards and cocaine itself. (Brain activations in general in response to a myriad of stimuli, such as cocaine cues, stress, MP, monetary rewards and cocaine itself correspond to the red and purple blocks. Brain activations of the limbic system in response to such stimuli correspond to the purple blocks. The limbic system is highlighted due its importance in substance use disorders in general. Laterality of the brain activations correspond to the blue and green blocks.)”

“Figure 3: Brain activations in the limbic system. (Brain activations of the limbic system in response to a myriad of stimuli, such as cocaine cues, stress, MP, monetary rewards and cocaine itself correspond to the purple blocks. The limbic system is highlighted due its importance in substance use disorders in general. Laterality of the brain activations correspond to the blue and green blocks.)”

Comment:

  1. 'Comprehensive' does not make sense for title of this study.

Response:

We have amended the title of the present manuscript to exclude such a terminology.

Comment:

  1. Main issue about this review article is lack of conclusive findings. The authors have mentioned several brain regions to be related with craving of cocaine without any conclusive analyses. Without a clear meta-analyses this findings is not so useful for future research.

Response:

Meta-analysis would be a major approach of user data synthesizing from different studies to get accurate conclusions. Unfortunately, the studies included in our review are highly heterogeneous in methodologies, outcomes, and populations, making it impossible to do a meta-analysis. Nevertheless, we hold the view that our review also remains useful in terms of summarizing and integrating information that currently exists on the neurobiology of the craving for cocaine. We emphasize that future studies must pay attention to the standardization of methods and outcomes so that the approaches of meta-analytic conclusions can be promoted. Measures of study designs and measurements smoothed out. The next meta-analyses will provide more clear and unambiguous results regarding the role of particular brain regions in cocaine craving. Additionally, we reinforce the fact that an in-depth investigation of this intricate subject will be vital in broadening our knowledge and the development of increasingly targeted interventions for those dealing with cocaine use disorders.

Reviewer 2 Report

Comments and Suggestions for Authors

Thank you for the opportunity to review this manuscript.

 In my opinion, it is an interesting paper, which obviously has limitations that were pointed out by the authors. One more limitation that was not taken into account was that the review included papers from more than a dozen years ago, while studies much more recent can be found in online resources. However, it may have been worth the effort to access papers archived by PsycINFO and Web of Science. 

The methodology seems clear, but it is a pity that the selection was done by one person. The error of subjectivity should be borne in mind here, but I cannot prove that this was the case. However, this is a bit of a weakness of this work. 

Instead, I like the discussion section, which was quite clearly described, even for laypeople and people who are not versed in brain function and dysfunctions caused by substance use. 

Drug (cocaine) starvation is an area of great interest. However, I have some doubts that the paper adds anything new to science. On the other hand, as a review article of other studies, it may be useful for those looking for condensed content in this area.

Author Response

We would like to extend our gratitude to the reviewer for their comprehensive feedback on our manuscript. We have carefully considered each point raised and made appropriate revisions to address the concerns raised. All the changes within the manuscript are highlighted in yellow. Below, we provide a detailed response to each of the reviewer's comments.

Comment:

In my opinion, it is an interesting paper, which obviously has limitations that were pointed out by the authors. One more limitation that was not taken into account was that the review included papers from more than a dozen years ago, while studies much more recent can be found in online resources. However, it may have been worth the effort to access papers archived by PsycINFO and Web of Science.

Response:

Thank you for your thoughtful critique. We appreciate your perspective on the potential value of accessing papers archived by PsycINFO and Web of Science. While we recognize the importance of these databases in providing access to high-quality scholarly literature, we made a deliberate decision to prioritize accessibility and comprehensiveness in our search strategy. As you rightly pointed out, PsycINFO and Web of Science are reputable databases known for their extensive coverage of psychological and multidisciplinary research, respectively. However, our decision to focus on PubMed, Google Scholar, Regional BVS Portal, and Scielo was motivated by several factors. Firstly, PubMed and Google Scholar were chosen for their broad coverage of biomedical and multidisciplinary literature, including both peer-reviewed journals and grey literature sources such as theses and conference papers. This enabled us to cast a wide net and capture diverse perspectives on the neurobiology of cocaine craving. Secondly, Regional BVS Portal and Scielo were included to enhance the inclusivity of our search by incorporating regional and non-English language sources, which may not be as extensively represented in PsycINFO and Web of Science. By expanding our search to these platforms, we aimed to capture a more diverse range of studies and perspectives on our topic of interest. While we acknowledge that our approach may have limitations, such as potentially missing out on some recent studies archived in PsycINFO and Web of Science, we believe that our comprehensive search strategy allowed us to mitigate this to some extent. Additionally, our focus on accessibility and inclusivity aligns with the principle of promoting diversity and representation in research. Nevertheless, your suggestion to consider accessing papers from PsycINFO and Web of Science is duly noted, and we will take it into account in future research endeavors. Thank you for highlighting this potential avenue for expanding the scope of our review.

Comment:

The methodology seems clear, but it is a pity that the selection was done by one person. The error of subjectivity should be borne in mind here, but I cannot prove that this was the case. However, this is a bit of a weakness of this work.

Response:

Thank you for your feedback regarding the methodology. We appreciate your concern regarding the potential for subjectivity in the selection process. To address this issue, it's important to note that all screening processes were meticulously reviewed by at least one additional author. When there was disagreement, a third author was consulted to reach a consensus. These details regarding the screening process and any discrepancies were thoroughly documented and are presented in Supplementary Tables 1 and 2 of the manuscript. We hope this provides assurance regarding the rigor and reliability of our selection procedure. We have clarified this process in the section 2.4 of the present manuscript, as follows:

“The first author conducted the entire article selection process in accordance with the Preferred Reporting Items for Systematic Reviews and Meta-Analyses (PRISMA). The flowchart illustrates the identification, screening, eligibility, and inclusion of articles, outlining the steps of this process leading to the final article selection (Figure 1). The records were screened by title and abstract (Supplementary Tables 1 and 2) by the first author and supervised by the last author. Disagreements were resolved by discussion with the second author.”

Comment:

Instead, I like the discussion section, which was quite clearly described, even for laypeople and people who are not versed in brain function and dysfunctions caused by substance use.

Response:

We appreciate your favorable review. For a variety of readers including those who are not acquainted with details of brain activities and chemical substance related dysfunctions, this serves the purpose of clarity and targeting the essence of the brain performance dilapidation. We were dedicated to give comprehensible explanations which will at the same time be eminently educational besides being engaging for both experts and non experts alike. We admit that in a panacea situation like this, we are making a lot of efforts, and we are really grateful to you for recognizing that.

Comment:

Drug (cocaine) starvation is an area of great interest. However, I have some doubts that the paper adds anything new to science. On the other hand, as a review article of other studies, it may be useful for those looking for condensed content in this area.

Response:

While we acknowledge that our review may not present entirely novel findings, it offers value as a comprehensive synthesis of existing literature on the topic. As you rightly pointed out, our manuscript serves as a review article to provide condensed content and insights gleaned from previous studies in the field. By compiling and analyzing existing research, we aimed to offer a comprehensive overview of the current state of knowledge regarding drug starvation, particularly focusing on cocaine. Our goal was to provide researchers and practitioners with a valuable resource that summarizes key findings, identifies trends, and highlights areas for further investigation. We are pleased that you recognize the potential utility of our review for individuals seeking condensed information in this area. While our paper may not introduce groundbreaking discoveries, we believe it contributes to the scientific discourse by synthesizing and contextualizing existing knowledge, thereby facilitating a deeper understanding of drug starvation and its implications.

Round 2

Reviewer 1 Report

Comments and Suggestions for Authors

Thank you for revisions.

Author Response

Thanks for such a careful review of our manuscript.